# 'I fit the category of the box, it just doesn't describe me well.' Exploring the perspectives of autistic women and gender-diverse individuals on self-report autism measures

Nora Uglik-Marucha[1]*, Serafina Show[1], Silia Vitoratou[1], Francesca Happé[2‡], Hannah Belcher[3‡]

1 Psychometrics and Measurement Lab, Biostatistics and Health Informatics Department, Institute of Psychiatry, Psychology and Neuroscience, King's College London, London, United Kingdom, 2 Social, Genetic and Developmental Psychiatry Centre, Institute of Psychiatry, Psychology and Neuroscience, King's College London, London, United Kingdom, 3 Health Service and Population Research, Institute of Psychiatry, Psychology and Neuroscience, King's College London, London, United Kingdom

‡ These authors are joint senior authors on this work.
* eleonora.uglik-marucha@kcl.ac.uk

## Abstract

Psychological assessments play a significant role in both clinical decision-making and the interpretation of research findings, with the quality of these inferences depending on the validity of the measures used. Recent evidence suggests there are gender differences in the presentation of autism, raising concerns about the validity of existing autism tools to measure autistic traits in women and the subsequent implications for clinical inferences and research. This study explored the perspectives of autistic women on the relevance of existing autism questionnaires to their lived experience, alongside additional input from gender-diverse individuals assigned female at birth (AFAB). Through interviews, focus groups, and online surveys, 22 autistic women and AFAB gender-diverse individuals shared their experiences using and perspectives on the Autism Spectrum Quotient-10, 14-item Ritvo Autism & Asperger Diagnostic Scale, and Broad Autism Phenotype Questionnaire. The interview data were analysed using reflexive thematic analysis, identifying two overarching themes: (1) questionnaires measure only one way to be autistic, and not in an autism-friendly manner, and (2) enhancing questionnaires' relevance for autistic women and individuals socialised as female: key missing experiences to include. The findings suggest that some of the most frequently used autism measures may not fully capture the experiences of autistic women and AFAB gender diverse individuals. Significant gaps were identified, indicating that important aspects of the participants' lived experiences were missing. Furthermore, concerns were raised about the questionnaires' lack of relevance to the autistic population as a whole. The findings underscore the non-satisfactory content validity of these tools for measuring autism in autistic women

**Data availability statement:** The data are not publicly available as the interview transcripts contain confidential, personal, and sensitive information that could compromise participant privacy, particularly given the inclusion of individuals from vulnerable communities, including those identifying as gender diverse. In accordance with the consent form, only disguised extracts may be used in the doctoral thesis and related scientific publications, and full transcripts cannot be shared. Requests for access to the data may be directed to Nora Uglik-Marucha (eleonora.uglik-marucha@kcl.ac.uk) or to the King's College London Research Ethics Committee (rec@kcl.ac.uk). All requests will be reviewed to ensure they comply with ethical and data governance requirements.

**Funding:** NU-M is funded by the NIHR [Doctoral Fellowship (NIHR302618)]. SV and FH are partially funded by the NIHR Maudsley Biomedical Research Centre at South London and Maudsley NHS Foundation Trust and King's College London. The funders had no role in study design, data collection and analysis, decision to publish, or preparation of the manuscript. No additional external funding was received for this study.

**Competing interests:** The authors declare that they have no competing interests.

and AFAB gender-diverse individuals. This highlights the need for their refinement to better reflect contemporary understandings of different presentations of autistic traits, particularly the impact of gendered experiences, in a way that avoids the introduction of possible new biases and remains relevant and accessible to autistic individuals.

## Introduction

Psychological assessments play a significant role in both clinical decision-making and the interpretation of research findings. The quality of these outcomes depends on the *validity* of the assessment measures used [1], that is, the extent to which an instrument measures the construct it was intended to measure [2]. Unlike physical properties that can be directly measured, psychological constructs, such as autistic traits discussed here, are not directly measurable and thus require indirect methods for assessment. It is crucial to ensure that psychological measures validly assess the constructs of interest in order to effectively test relevant theories and draw accurate inferences. Notably, even the most advanced statistical methods cannot rectify issues arising from construct validity shortcomings [3,4], which then shape current knowledge and practice.

Recently, concerns have grown regarding the validity of psychological assessments, with evidence for their construct validity often found to be lacking or insufficient [5–8]. Schimmack [7] described this issue as a validation crisis in psychology, urging a comprehensive investigation into the validity of widely used psychological measures. This issue also extends to autism research, where scholars (e.g., Livingston et al. [9], Schiltz et al. [10], and Williams [11]) have highlighted issues with the validity of instruments used in this field. Given the dynamic and evolving nature of instrument validity, which can deteriorate over time due to evolving theories and constructs [1], and the substantial changes in the concept of autism over recent decades, these concerns are warranted. Autism has been reconceptualised from a narrow and rare diagnosis to a broader, relatively common condition; from a purely childhood-limited condition, to a wider lifespan perspective [12]. The concept of autism has further shifted from being a unitary, distinct diagnostic category with a single underlying cause to one that acknowledges the presence of autistic traits beyond diagnostic boundaries, multiple underlying aetiologies, and co-occurrence with other conditions [12]. This shift has been also reflected in a move away from a purely medical model towards a framework that increasingly embraces neurodiversity [12].

Males have historically been diagnosed as autistic more frequently than females, with most commonly reported estimates showing a male-to-female ratio of approximately 3–4:1 [13–15]. More recent global estimates suggest this ratio is closer to 2:1 [16]. Indeed, recent years have seen an increase in diagnoses among girls and women, though often later in life [17,18], and underdiagnosis still remains a significant issue, particularly in adulthood [19]. One proposed explanation for the gender disparity is the emerging evidence that autistic traits in women and girls may present in subtler ways than those described in traditional diagnostic criteria [20]. Concerns

have consequently been raised that existing autism assessments may not fully capture autistic traits more often observed in females, potentially compromising their validity for measuring autism in women and girls [21–23]. It is crucial to note that although those subtler variations of autistic traits might occur more frequently in autistic women on average, these traits are not strictly gender-specific and individuals of other genders may also exhibit this presentation. This is especially relevant given the higher proportion of autistic individuals identifying as gender diverse [24], particularly those assigned female at birth (AFAB) [25].

Lai et al. [21] identified three levels related to the measurement of autism. The first level refers to 'broad constructs,' which correspond to the two main domains of autism, namely social communication differences and restricted and repetitive behaviours and interests (RRBIs), at an abstract level. Second, 'narrow constructs' represent subdomains within the two main domains. This could include restricted interests under RRBIs, or differences in developing, maintaining, and understanding relationships within the social communication domain. The third level relates to 'behavioural exemplars,' which encompass specific behaviours expressing a narrow construct, such as the type of restricted interest. Lai et al. [21] suggested that gender differences in autistic traits should be discussed primarily at the levels of 'narrow constructs' and 'behavioural exemplars', a view further supported by Cook et al. [26], who proposed that these differences mostly manifest at the 'behavioural exemplars' level. Recent systematic reviews and meta-analyses at the *narrow* construct level indicate that autistic females tend to exhibit better social communication skills than autistic males [27] and that autistic males show higher levels of RRBIs, specifically in stereotyped behaviours and restricted interests, with no gender differences found for insistence on sameness and sensory experiences [28]. However, findings become more inconsistent and complex when also considering meta-analyses at *broad* construct level. Most meta-analyses at this broader level suggest that autistic females, on average, display fewer RRBIs than their male counterparts [29,30], although a recent meta-analysis found no gender differences [31]. Likewise, while earlier broad-level meta-analyses reported no gender differences in social communication [29,30,32], more recent work, which evaluated social communication as two broad constituents separately, suggests that autistic females, on average, demonstrate better social interaction skills but no differences in communication [31].

In line with the recommendations by Lai et al. [21], the following discussion elaborates on findings from meta-analyses regarding *narrow-level* autistic traits; however, further research at the *narrow-construct level* remains necessary. Better social communication skills in autistic females may be reflected, for instance, by displaying higher social motivation to seek friendships and social interactions compared to autistic males [33,34], despite facing challenges in forming and maintaining friendships, as well as resolving conflicts within them [34–37]. Better social communication skills could also be explained by masking, that is, the use of conscious or unconscious strategies to change social behaviour so that autistic differences are less apparent [38,39] or compensated for [40], in order to facilitate navigation of predominantly neurotypical social contexts [41]. Although aspects of masking likely extend to non-autistic people as well [42,43], for autistic people it can come with distinct motives and costs [44], and may be modulated by gender [45], potentially contributing to the under-recognition of autism in females. In terms of RRBIs, gender differences may be qualitatively reflected in the types of interests reported: autistic females are more likely to engage in relational interests, such as those involving people, nature, art, and psychology, while autistic males are more likely to have interests centred around mechanical themes, including technology, gaming, or vehicles [46–49]. However, existing autism measures may overlook the presence of autistic interests and social communication differences in females due to the 'gender-typicality' of the topics of interest [50] and the higher social motivation often observed in autistic females while potentially not capturing more nuanced social interaction challenges and not accounting for masking.

In light of the substantial changes in the conceptualisation of autism and the emerging evidence of its diverse presentations, we adhere to Haynes et al.'s [1] recommendation for the continuous review and updating of content validity of frequently used psychological measures, which constitutes a crucial aspect of construct validation [51]. According to Haynes et al. [1], content validity is *the degree to which elements of an assessment instrument are relevant to and representative*

*of the targeted construct for a particular assessment purpose*. While often associated with item development, content validity applies to all aspects of a questionnaire that affect the data collected, including item content, instructions, response scales, and scoring. Haynes further elaborates that content validity involves ensuring that a measure (i) includes items that are essential and representative of the construct, (ii) excludes items that assess constructs outside the target domain, and (iii) maintains an overall score that is not disproportionately influenced by any single facet of the construct. This underscores that content validity concerns the entire scale, not just its individual items [52]. It is important to note that content validity is also conditional and thus may not necessarily generalise across different contexts. It can depend on its intended use, that is, a measure may be content valid for one specific purpose, such as screening, but not necessarily for other purposes, such as evaluating treatment effectiveness, because the inferences drawn from an instrument can vary depending on its function [1]. Similarly, content validity can be conditional on the target population [52–54], meaning that a measure can be valid for one group but not for another, and therefore it is important to ensure that content validity of a measure is established in the specific target group in which the tool will be used. In the context of autism assessments, a self-report questionnaire of autistic traits would exhibit satisfactory content validity, first, if it encompasses items covering all aspects of the autism construct as defined by the seven subdomains of diagnostic criteria in the DSM-5 [55] or contemporary theories of autism (for instance, monotropism [56]). This could also include additional facets of the autism construct identified as important by clinical, academic and lived-experience experts and theory [10,57]. Second, it would exclude items that assess unrelated constructs, such as anxiety or depression. Third, it would proportionally represent each subdomain of autism, with a balanced number of items for each subdomain and appropriate weighting in the aggregate score. However, the development of these items must be guided by the measure's intended function and target population to ensure its content validity is satisfactory. Failing to meet the above standards can lead to inferences that overrepresent, omit, or underrepresent some facets of the construct and/or reflect variables outside the construct domain. This can in turn affect latent variable models, internal consistency of a measure, causal models, predictions, participant selection in research, treatment effect estimates and diagnosis [1,58].

The evaluation of the content validity of a measure can include both quantitative and qualitative approaches. Quantitative methods often involve expert panels evaluating the relevance, clarity, and representativeness of a measure's elements, with their ratings quantified through content validity estimates such as the content validity ratio [59], content validity index, modified kappa [60], formal content validity analysis [61], Bayesian formal content validity analysis [61], among others (for summaries of these methods, see Spoto et al. [61], Almanasreh et al. [58], and Colquitt et al. [62]). Qualitative methods, such as interviews and focus groups with experts and individuals from the target population (experts by experience), are particularly valuable to not only ensure that the items are relevant and representative of the construct but also to identify missing aspects that are essential to fully capturing the intended construct and highlighting areas where construct refinement may be needed [1,57]. Pretesting questionnaires using these qualitative methods can help to ensure robust item and measure construction, which is crucial for achieving high-quality, unbiased measurement [4]. Such pretesting could identify potential sources of group bias in measurement and contribute to understanding how group membership might influence response processes and question interpretation [4]. Detecting these issues early in questionnaire development allows for adjustments to ensure that the questionnaire accurately measures the same construct across different groups.

Some qualitative studies involving autistic individuals have already raised concerns about the content validity of diagnostic tools [63–65] and questionnaires [66–68] to measure autistic experiences in a way that is relevant to autistic people themselves. One study [64] highlighted issues with the relevance of play behaviour assessments in diagnostic tools for autistic women, noting that such assessments can overlook the impact of masking or the presence of play styles that resemble those of non-autistic individuals. Another study [65], which specifically focused on the experiences of autistic women during diagnostic assessments in the United Kingdom (UK), reported that diagnostic tools were perceived as tailored to the 'male' presentation of autism and assessment in children. The assessment process was also seen as lacking

questions about traits and experiences more common in autistic women, particularly those related to autistic passions, stimming, masking, menstruation, sex, menopause, and experiences of sexual abuse or exploitation. To date, only two studies have specifically examined the relevance of self-report questionnaires of autistic traits to autistic women [67,68]. One mixed-methods study [67] has investigated this in relation to the Autism Spectrum Quotient-50 (AQ-50) [69], the Girls Questionnaire for Autism Spectrum Conditions (GQ-ASC) [70], and the Camouflaging Autistic Traits Questionnaire (CAT-Q) [71], utilising both close-ended questions and an optional open-ended comment to gather data. The qualitative findings indicated that autistic women viewed the AQ-50 as reinforcing a stereotypical view of autism, which does not align with their autistic experience and is particularly unsuitable for autistic women who can mask. The CAT-Q was noted for its repetitiveness and difficulty in comprehension, with participants noting the necessity of a high level of self-awareness to respond to items. Meanwhile, the GQ-ASC was deemed inappropriate for adults or those diagnosed later in life, with specific issues identified regarding items that required respondents to compare their behaviour to other girls. Despite these drawbacks, the GQ-ASC was preferred over the other two measures. A second study [68] used interviews to explore the perspectives of Chinese autistic females towards the AQ-50. The authors highlighted that the lived experience described by autistic women challenged the stereotypical descriptions of autism embedded in the measure, which also overlooked experiences relevant to autistic women, such as empathy, verbal abilities, and understanding of social conventions. Several items were described as outdated in the context of a technologically advanced society and difficult to interpret, partly due to translation issues.

While there has been a growing body of research in recent years focusing on the experiences of autistic women and gender differences in autistic traits, some of which have raised concerns about potential gender bias in measurement [21–23], there is a scarcity of qualitative studies exploring autistic women's perspectives on the relevance of autism questionnaires to their lived experience. This is one of the first studies to address this gap by conducting in-depth interviews with autistic women to evaluate their perspectives on three widely used self-report autism measures, namely the Autism Quotient-10 (AQ-10) [72], Ritvo Autism & Asperger Diagnostic Scale (RAADS-14) [73], and Broad Autism Phenotype Questionnaire (BAPQ) [74]. We also welcomed and included views of AFAB gender-diverse individuals who considered their experiences relevant to the study's aims. In keeping with participants' own terminology, we use the term 'socialised as female' interchangeably with 'AFAB gender-diverse' throughout the manuscript. By capturing these perspectives, this study seeks to understand whether the conceptualisation of autism as reflected in these questionnaire items aligns with the experiences of autistic women and AFAB gender diverse individuals. This will help to ascertain the potential presence of gender-bias in these measures and its sources, and to identify any additional autism-related constructs that should be included to make these measures more relevant to this population.

## Methods

### Procedure

Ethical approval was obtained from King's College London Psychiatry, Nursing & Midwifery Research Ethics Subcommittee (REC Reference Number: LRS/DP-22/23/34273). The Qualtrics platform was used to provide written and video information on the study, obtaining informed consent, screening, collecting demographic information, capturing participants' preferences for interview modalities and contact information. Capacity to provide consent was ensured by excluding individuals under 18 years of age and those with a diagnosis of severe intellectual disability or severe learning difficulty. For all other participants meeting eligibility criteria, capacity was assumed. All survey materials and the interview guide were developed in collaboration with autistic adults recruited through the Autistica Insight Group. Autistic collaborators received compensation of £25 voucher per hour for reviewing the materials or study participation.

Participants were recruited through the Autistica charity, the online research platform Prolific, and social media sites, namely X (formerly Twitter) and Reddit. The study was advertised as research exploring the perspectives of autistic

women and gender-diverse individuals on the relevance of existing autism questionnaires, with the aim of informing the development of a measure better suited to identifying autistic traits within this population. Autistic adults were eligible to participate in the study if they (1) self-identified or were diagnosed as autistic; (2) identified as (cis/trans) women; however, perspectives of individuals AFAB but not identifying as women were also welcomed; (3) were above 18; (4) fluent in English; and (5) had no diagnosis of severe intellectual disability and/or severe learning difficulty. Participants self-identifying as autistic were encouraged to participate acknowledging the underdiagnosis, misdiagnosis, or late diagnosis of women [75], as well as the existing barriers faced by autistic individuals, particularly women [76,77], in accessing a diagnosis [78,79]. To ensure the experiences of underrepresented communities in autism research were included [80,81], participants from ethnic minorities were prioritised for interview invitations, and participants from Black ethnic backgrounds were additionally recruited through the Prolific service to address a low initial interest from this group.

## Interviews

An initial semi-structured interview guide was developed by the research team and subsequently reviewed by four autistic women and one autistic individual who identified as genderfluid. Their feedback informed refinements to the guide, such as the addition of the question, 'Were there any questions you think were missing from these questionnaires?', which resulted in the final version used in this study. Semi-structured interviews were selected to maintain a balance between providing a structure and allowing flexibility. This format facilitated additional comments while ensuring coverage of key areas of interest with all participants. Please see S1 Table for the interview guide.

Different modalities for the interview were offered to participants, including individual and focus group Zoom interviews, as well as an online Qualtrics survey for written responses. 12 (54.55%) participants chose one-on-one interviews, 7 (31.82%) opted for the online survey, and 3 (13.64%) participated in a focus group. Online interviews offer the advantage of face-to-face communication, while written communication can provide autistic participants with a more structured and predictable mode of communication, reducing anxiety and generally being preferred by autistic individuals [82]. A week before the scheduled interview, participants were provided with the interview questions and relevant measures (please refer to the Measures section) that would be discussed during the interview. NUM and SS conducted the interviews. Participants were offered the opportunity to meet with the interviewers prior to the interviews, with one participant choosing to do so. The interviews were audio recorded and transcribed verbatim. Survey responses, already in text format, were directly copied into a data file. Participants were asked to provide a pseudonym for quoting purposes, used below when referencing quotes.

The decision to conclude interviews and finalise the sample size was guided by the concept of information power [83], which considers the richness of the data in relation to the study's aim, sample specificity, use of established theory, quality of dialogue, and analytic strategy. The study aimed to explore perspectives broadly, focusing on autistic women and those socialised as female, while ensuring variation in diagnostic journeys, age, and representation from underrepresented communities. This required a larger number of participants to capture sufficient diversity across diagnostic status, age, race, and gender diversity. The use of multiple modes of communication also necessitated additional participants, as preferences for specific modes could reflect additional diversity. For example, less verbal participants may have opted for written responses, requiring sufficient representation within each mode for this to be explored. Considering these factors during data collection and familiarisation, it was concluded by the 22nd interview that the study had achieved satisfactory information power to explore the research question in-depth.

## Measures

Participants were asked about their perceptions and experiences related to three selected measures of autistic traits, namely AQ-10 [72], RAADS-14 [73], and BAPQ [74]. These measures were selected based on several criteria. First,

their extensive use in either research, each cited over 150 times on Google Scholar, or clinical practice; that is, they were recommended by the National Institute for Health and Care Excellence (NICE) [84] for autism screening or assessment in adults. Preference was given to shorter versions where available to reduce participant burden. For example, although the Ritvo Autism Asperger Diagnostic Scale-Revised [85] is recommended by NICE [84], its abbreviated form, the RAADS-14, was selected for brevity. Second, each measure was developed to assess a broad range of autistic traits across diagnostic criteria, rather than focusing on one specific domain of autism. Finally, all three measures were validated for use in adult populations regardless of gender and were available as open-access measures.

**AQ-10** [72] is a brief self-report measure consisting of 10 items that assess autistic traits on a four-point scale, ranging from 1 (definitely agree) to 4 (definitely disagree). For scoring, 'definitely agree' and 'slightly agree' are scored as 0, while 'definitely disagree' and 'slightly disagree' are scored as 1 for reverse-scored items, with the opposite applied to non-reverse-scored ones. This questionnaire is a shorter version of the Autism Spectrum Quotient-50 [69] and is recommended by NICE [84] to screen for autism in adults. Widely utilised in research, it has accumulated over 800 citations on Google Scholar and is commonly employed for participant inclusion and exclusion criteria in studies. Nevertheless, despite its extensive use in both research and clinical settings, psychometric concerns have been raised regarding the measure [86,87], warranting further evaluation and caution in its continued use.

**RAADS-14** [73] is a 14-item self-report measure of autism, which asks to rate the presence of autistic traits across the lifespan using the options: 3 (true now and when I was young), 2 (true only now), 1 (true only when I was younger than 16), and 0 (never true). It is a shorter version of the original 80-item Ritvo Autism Asperger's Diagnostic Scale-Revised [85], which is recommended by NICE [84] for use in the assessment of autism in adults. The RAADS-14 is well-received in the literature, evidenced by over 150 citations on Google Scholar.

**BAPQ** [74] is a 36-item measure available in both self-report and informant-report versions. The questionnaire was designed to assess 'milder' characteristics of autistic traits that share qualitative similarities with the main components of autism, commonly seen in relatives of autistic people, referred to as the broader autism phenotype. The measure asks participants to rate how frequently each item is applicable to them, ranging from 1 (very rarely) to 6 (very often). BAPQ is widely recognised in research, accumulating over 180 citations on Google Scholar.

## Data analysis

The data were approached with the aim to understand perspectives and experiences of autistic women and those socialised as female regarding existing questionnaires for autism. Reflexive thematic analysis (RTA) [88] was chosen as the most suitable method, offering inductive exploration of patterns of perspectives within the data to identify themes. The theoretical flexibility of RTA allowed it to be informed by the neurodiversity paradigm, emphasising the legitimacy of autistic voices in describing their own perspectives while also situating them within broader sociocultural contexts.

The first author led the analysis, beginning with immersion in the data through repeated listening to recordings and re-reading of transcripts. Initial coding labels were generated in NVivo 13 for MacOS and refined through two additional rounds of recoding. Themes were developed by grouping codes with shared meanings, creating a directory of candidate themes and subthemes with definitions and supporting quotations. These themes were further refined through discussions with the research team. A near-final draft of the manuscript was shared with two study participants for feedback on language used when referring to participants, theme naming, and overall findings, for which they were reimbursed. The writing process played a key role in shaping the analysis and finalising the theme structure [88].

## Positionality

The authors continuously reflected on their positionality throughout the study process, recognising that researchers' experiences, values, and motivations shape decisions and interpretations at every stage of the study [89]. At the time of

interviews and data analysis, the first author self-identified as autistic and later received an official diagnosis. However, this self-identification was not disclosed to participants, which may have set a precedent for different communicational styles and shaped participant responses. This study was motivated by the first author's postgraduate dissertation on autism, where participant feedback highlighted the limitations of commonly used autism measures. This experience underscored the importance of community consultation and improving measures to better reflect autistic experiences, particularly for those with more subtle variations of autistic traits, largely driven by the researcher's personal experience. Although the first author's training and expertise lie in psychometrics, and they therefore inherently viewed the data through a quantitative lens, they remained conscious of it, striving to embrace subjectivity in the analysis.

The remaining research team comprises both neurodivergent and neurotypical researchers who identify as women or gender diverse and align with the conceptualisation of autism within the neurodiversity paradigm.

## Results

### Participants' characteristics

Twenty-five individuals initially participated in the study; however, 3 participants were excluded from the analysis due to scamming activity. These individuals were identified as scammers based on patterns observed in previous reports on fraudulent participants [90], including very brief and vague responses, a lack of familiarity with autistic experiences, and high fraudulent scores flagged during post-interview screening in Qualtrics. The final sample comprised 22 participants, of whom 16 (72.7%) self-reported a diagnosis of autism and 6 (27.3%) self-identified as autistic. The majority of participants (n = 17, 77.3%) identified as women, 2 (9.1%) as non-binary, 1 (4.55%) as genderqueer, 1 (4.55%) as agender, and 1 (4.55%) identified as other. For ethnicity, 7 (31.8%) identified as White/Caucasian, 6 (27.3%) as Asian/Asian British, 6 (27.3%) as Black/African/Caribbean/Black British, and 3 (13.6%) as belonging to mixed/multiple ethnic groups. The participants had a mean age of 30.82 years (SD = 8.97) ranging from 18 to 60 years old, with half having a university education. Additionally, more than half of the sample reported having neurodivergence other than autism, and the majority endorsed having co-occurring mental health conditions. Detailed demographics are available in S2 Table.

### Overview of the themes

Participants' perspectives and experiences of items from existing autism questionnaires in relation to their own experiences of autism are outlined in two overarching themes presented with subthemes in Fig 1.

### Theme 1: Questionnaires measure only one way to be autistic, and not in an autism-friendly manner

**Subtheme 1.1: Stereotypical presentation of autism.** Participants expressed that the questionnaires appeared to emphasise a narrow set of stereotypical autistic traits commonly associated with autism as a male childhood condition, conceptualised within a medical model of disability. They highlighted that the questionnaires mainly focused on describing a presentation of autistic traits that is visibly noticeable to others, as typically seen in those identified as autistic in childhood, especially boys. These traits are frequently characterised by external behaviours that others may view as problematic or indicative of distress.

*It's kind of stereotypical. It's kind of giving an energy where, oh, there's this like autistic boy who has sensory issues, very obvious sensory issues, and then like, you know, some amount of trouble communicating with people. (Helen, woman, aged 25)*

Aubrey provided an example of how the stereotypical perception of autism as a predominantly male childhood condition is reflected in the questionnaire items.

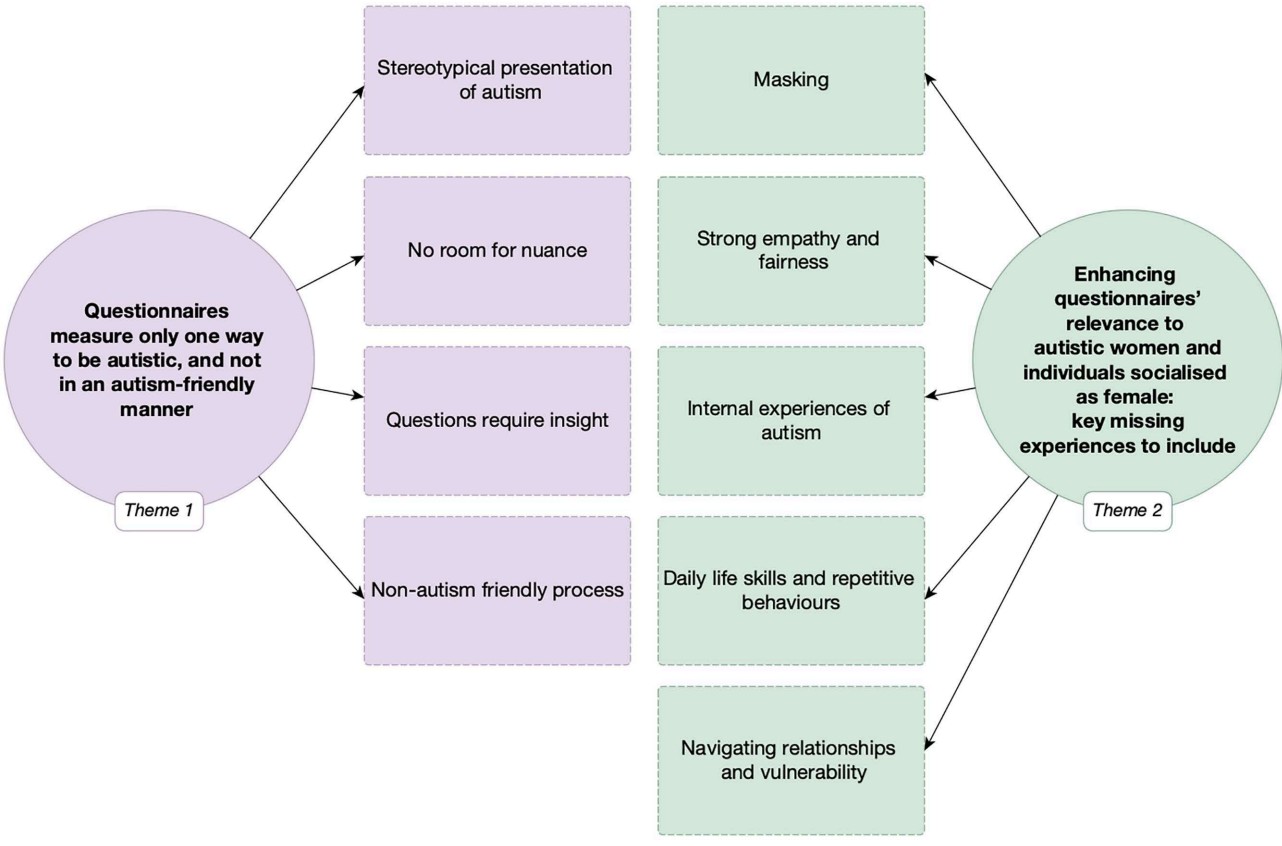

**Fig 1. Overview of the themes and subthemes.**

*I mean, 'I like to collect information about categories of things' like, you know, that's a very kind of male little boy stereo-typed picture of autism, I think. And for me, as like a middle-aged woman, I'm just like, well, no, I don't collect information about trains. So, yeah, I find it a bit weird to answer that as like a female adult. (Aubrey, woman, aged 48)*

Questionnaire items based on such conceptualisation of autism often felt irrelevant to autistic women or those AFAB, who may have interests in areas considered more 'normative,' such as celebrities and animals *(Sandra, woman, aged 30)*, relational games like The Sims (*Poppy, non-binary, aged 29*), fashion, makeup, art (*Gemma, woman, aged 22*), or subjects like psychology, sociology, and philosophy (*Robin, woman and non-binary, aged 30*).

Describing autistic traits through observable behaviours was recognised by participants as potentially unsuitable for autistic women or individuals socialised as female. Some pointed out the influence of gendered socialisation, noting that women or those socialised as female may feel greater pressure to mask their autistic traits and thus may not be perceived as visibly autistic. Kim discussed their struggles with responding to some questions, as they tended to internalise their autistic experiences rather than express them outwardly.

*I found when I've spoken to other women or non-binary people, or just anyone female presenting, things like struggling to focus in social situations, and not knowing how to function as part of social groups, that we struggle with it but we don't outwardly show it. So then it's, how do I, which box do I tick with that? (Kim, non-binary, aged 28)*

While several participants noted that the measures focus on outward autistic traits typically associated with a 'male autism' stereotype, rather than internal experiences that may be more common among autistic women or those socialised as female, Poppy cautioned against framing these traits strictly in terms of gender:

*I'd be wary of saying it's a gender split. I think it's more that there's an externalised presentation that is very easy to spot; more likely to be seen in young boys and those who are identified as being autistic when they are children and this more subtle internalised presentation that isn't as obvious and isn't as annoying to the grownups. (Poppy, non-binary, aged 29)*

Participants also expressed their feelings around the questionnaires' focus on deficits, which they felt reinforced a stereotypical perception of autism as inherently pathological. This led to feelings of judgment and, in some cases, offense. This perspective is further emphasised by Ibiode's impression of autism as portrayed in these questionnaires:

*I feel like sometimes they [the screening questionnaires] make you feel like you can't have a partner, you can't be doing well in work. It has to be awful. Like your life has to be awful for you to be autistic. (Ibiode, woman, aged 24)*

Fatin concluded that:

*It would be a bit unfortunate if this is how people were learning about their own autism, just because it is so deficit focused. (Fatin, woman, aged 27)*

**Subtheme 1.2: No room for nuance.** Participants emphasised the lack of nuance in the questions, noting that they were either overly specific or overly general. Robin explained that overly specific statements may lead those who do not experience that exact expression of the trait to perceive the statements as not applicable, even though they could be relevant in a broader context.

*[In response to: I have to cover my ears to block out painful noises] I think some autistic people would say, well, yeah, I do cover my ears, but I don't consider the noise as painful. I consider the noise to be something else. And so they might say, well, [the question]'s not appropriate for me. (Robin, woman and non-binary, aged 30)*

Building upon Robin's point, Emilia highlighted that overly specific statements may not be suitable for autistic individuals because they may interpret them literally:

*We'll take the question as what you've asked, not all the hidden meaning that might be behind it. (Emilia, woman, aged 32)*

The specificity of statements related to sensory sensitivities was highlighted, particularly their overemphasis on sound hypersensitivities (for example, '*why have they chosen sounds? Why not another sensory thing?*' [Robin, woman and non-binary, aged 30]). They wished for more acknowledgment that sensory experiences can vary among individuals (for instance, '*that's only referring to one sensory input or output, it's about recognising that people's sensory sensitivities are different*' [Rose, woman, aged 31]), and that autistic people can experience not only hypersensitivity but also hyposensitivity to their sensory environment (for example, '*that [question] assumes that people are hypersensitive to things they might not be*' [Aubrey, woman, aged 48]).

The second concern raised by participants was that some questions were deemed overly general. Aubrey specifically noted how such statements clash with the very nature of being autistic:

*If you're assuming that autistic people are very focused on the details, and they find it difficult to generalise, why ask questions that are about a very general experience. (Aubrey, woman, aged 48)*

Participants felt that these statements failed to account for contextual variations in behaviour. For example, Emilia noted that the interactions with others could greatly vary depending on several factors:

*'I like being around other people'. And what I've written down is all the questions that came into my brain in relation to how to answer that, which was: who are these people? When? Where? For how long? What are we doing? Are there hundreds of them? Can I hear myself think? Can I leave? […] So like that, I have no idea how to answer that question. (Emilia, woman, aged 32)*

**Subtheme 1.3: Questions require insight.** Participants underscored concerns regarding questions that ask about one's experiences in relation to others, noting that such statements assume self-insight into how one's behaviours or perceptions compare to those of different people. Skye shared her experience with such questions:

*I always kind of assume that my experience is the same as other people's […] like the very first one in AQ10, 'I've noticed more sounds when others do not.' I assume that I'm hearing the same amount of small noises as everyone else. (Skye, woman, aged 36)*

This can be further exacerbated when the statements do not take into account the immediate social environment of autistic people, such as family or friends who might also be neurodivergent, potentially distorting the perception of 'typical' behaviours or experiences. Daisy recalls their experience of this, noting that they initially perceived some autistic traits as commonplace:

*My Dad is autistic as are several of my siblings, and my Mum has ADHD. They were all undiagnosed throughout my childhood, and as such my view of a typical family life was skewed. I thought echolalia, stimming, and extreme sensitivity to touch/taste/sound, etc., were normal. When I have been asked 'do you notice sounds around you that others don't', the answer is no, because my whole family also notice those sounds, as do my autistic friends. (Daisy, agender, aged 30)*

Some statements were also perceived as expecting insight into other people's feelings and thoughts through reading body language, which was described as unfair for autistic people to answer, who may not be able to recognise such cues as effectively. Participants expressed uncertainty about knowing if they accurately interpreted these cues, while others mentioned that the statements do not consider the possibility that one might falsely believe they interpreted something correctly.

*'I find it easy to work out what someone is thinking or feeling just by looking at their face' […] I feel like I do, but how can you actually know, right? Because you can feel like you're very good at it, but you're making all these wrong decisions. (Ibiode, woman, aged 24)*

Some participants also reflected on the challenge of answering statements while considering the impact of masking. They noted that recognising that one is masking and the difference between their masked and unmasked selves requires a deeper level of insight, as explained by Gemma:

*[…] if you have been bullied a lot and you've learned to mask as a coping mechanism and you don't know that you yourself are masking if you fill this out, you might give incorrect answers because... [you] don't have that self-awareness to have identified the fact that you are in fact masking. (Gemma, woman, aged 22)*

Questions were observed to demand not only insight into oneself but also into autism itself. Participants described how their experience of completing these questionnaires was different depending on whether they had prior knowledge about

autism and how their answers changed once they learnt what the questions were testing for. Sarah reflected on revisiting their responses from the assessment:

*I ended up like looking back at my responses. And I, now that I have more insight and information, like there were things that I answered 'never' that were not true. (Sarah, genderqueer, aged 36)*

**Subtheme 1.4: Non-autism friendly process.** Participants noted that completing questionnaires can be mentally exhausting for autistic people as they tend to be more detail-oriented. This can involve analysing questions very meticulously and consulting their close friends/relatives to ensure they provide precise and accurate answers, in contrast to neurotypical individuals who may take a quicker and less deliberative approach.

*[…] I think most neurotypicals would whip through it, find it really quick and easy, not bother phoning their mum, their dad, et cetera. I think they'd just be there going, 'yep, no, I'm sure nobody does that, or maybe a bit, or no, actually, no.' And I reckon they'd be done in 10 minutes. And certainly, when I was doing my autism assessments, I spent ages doing them and going back and checking and checking with others and checking that I wasn't sort of misrepresenting myself. (Emilia, woman, aged 32)*

Others wished for someone to clarify the meaning of the statements to ensure they were not misinterpreting themselves. Kim recalled experiencing this during their screening process for an autism assessment:

*I don't know, sometimes I think it's just more me overanalysing each of the little bits. And maybe, I don't know, maybe if GPs had the time and the headspace to go through these questionnaires with people, then the responses to the questionnaires might be more accurate. There's a lot that I left blank and put little caveats of 'I'm not 100% sure what you'd like from me. I think this is the answer, but here you go.' (Kim, non-binary, aged 28)*

The inability to provide accurate responses was especially noticeable in the response scales of the questionnaires, where the lack of clear quantitative distinctions between response categories could lead to frustration:

*And I can't tell you the distinction between like 'occasionally' and 'rarely' and 'somewhat often', those all kind of sound the same. […] I think I'd be so focused on making sure I was giving an appropriate answer that was actually true that like, that could have stressed me out. (Sarah, genderqueer, aged 36)*

**Theme 2: Enhancing questionnaires' relevance to autistic women and individuals socialised as female: Key missing experiences to include**

**Subtheme 2.1: Masking.** Participants noted that crucial experiences specific to autistic women and those socialised as female were missing from the questionnaires. They suggested including items about masking to better address their specific experiences. Participants emphasised that women or those socialised as female may often encounter greater pressure to conform to social conventions compared to men, with society penalising them more severely for social mistakes. They pointed out that this pressure can be amplified for autistic women, who may be particularly aware of their differences in adhering to social norms. As a result, autistic women may engage in masking behaviours more than autistic men, which is a crucial aspect that should be reflected in the questionnaires to enhance their relevance to autistic women and those socialised as female.

*I think females are socialised in a different way from males and I think that has a huge impact. So a lot of the questions I think are badly worded for people who've been socialised in a female type way. And so one of the things that I thought*

*would be really helpful would be to ask questions about things you'd rather do or wish you could do rather than what you actually do and what you actually might be doing is because of rules and expectations. (Emilia, woman, aged 32)*

Participants particularly referenced statements pertaining to social interaction, small talk, and group work, noting that while they can mask to perform these behaviours, there are no statements that would account for the substantial effort, practice, and self-teaching needed to develop them. Sarah summarised this perspective:

*I can chit chat. Yes, I can do it. I hate it. But like I can. It's just not something I ever want to do. And it certainly isn't inherent. (Sarah, genderqueer, aged 36)*

Maeve emphasised that items about masking need to consider that individuals might not be aware of the concept of masking or recognise that they are in fact doing it:

*But people might not know what masking is or that they do it, especially if they're filling in an autism screening questionnaire and don't necessarily know everything about autism yet. I think there should be questions which establish whether someone is masking, e.g., 'do you feel like you have to put on a fake smile when dealing with new people.' (Maeve, woman, aged 26)*

Building on this point, Ibiode emphasised that questions about masking should take into account that directly asking whether someone engages in specific masking behaviours may not be ideal. By adulthood, masking can become so ingrained that it can feel like an integral part of a person's identity. This integration makes it challenging to differentiate between masking behaviours and their 'true' self, as masking becomes an automatic behaviour.

*[…] So there should be questions about masking, but it needs to be in a way that's not...You're pretending to be someone else. Once you've got to be an adult, it kind of forms part of you a little bit. Like, you know, it's not just going out and try[ing] to not pretend. It would be very hard to do that. (Ibiode, woman, aged 24)*

However, some participants pointed out that the response scale of RAADS-14, which compares the presence of autistic traits currently versus in the past, could potentially address these concerns. Ruth explained how she found it beneficial for recognising and understanding the development of masking behaviours at different stages of life.

*I think the RAADS way of doing it, that 'did you have issues when you were younger, do you have issues now,' is a more interesting way of doing it. Because at the end of the day, that more shows your well, your ability, my ability, anybody's ability to mask and how that can change. I found that up to the age of 17, a bit of an arbitrary figure. Because if you're female and you learn to mask, you're doing it way before then. Way, way before then. (Ruth, woman, aged 41)*

**Subtheme 2.2: Internal experiences of autism.** Participants emphasised the need for questions that explore the internal experiences of autism, as current questionnaires tend to focus on external traits. This was identified as crucial because individuals might not be aware of their own autistic behaviours, and autistic women, in particular, may not outwardly appear autistic due to masking. Such questions could help identify those who internalise their autistic traits or have learnt how to navigate situations in a neurotypical manner, even if it does not feel natural to them.

*...Perhaps we need more questions that are focused on how we feel rather than how we behave, because how we behave, A, isn't necessarily that known to the autistic person because they don't have a mirror. And B, is so powerfully affected by masking, which is so powerfully affected by gender. (Robin, woman and non-binary, aged 30)*

Emilia emphasised that such questions should specifically delve into the reasons behind autistic traits, rather than merely noting their presence. She suggested that understanding what one feels and the function of a specific autistic trait is crucial, as simply identifying the presence of behaviours only scratches the surface of what autism is.

*But they absolutely don't cover why and to me, the why is what makes me autistic. It's not the eating the same food, because lots of people do that. Lots of people do that out of laziness or because it's cheap or it stops their ham going off. But I think the why is what makes us autistic. It is having these kind of systems and rules and easier ways of doing things. I think we are very systematic, generally. So I think quite a lot of the questions don't cover that very well. (Emilia, woman, aged 32)*

Participants frequently noted the lack of questions addressing internal experiences, particularly concerning internalised meltdowns or distress. They noted that socialisation often teaches women and those socialised as female to prioritise others' needs and suppress their own feelings, which can lead to internalising distress rather than displaying it publicly. Some participants also highlighted that they self-harm when overwhelmed by non-autism-friendly environments or masking, which is rarely recognised as a form of meltdown. They felt this tendency was more common among women or those socialised as female and suggested that including these aspects would make the questionnaires more relevant and validate their experiences. The current focus on external behaviours, they argued, implies that only observable behaviours are recognised as legitimately autistic.

*Maybe the questionnaire could talk or ask more about being overwhelmed, because Poppy put [in chat], 'I self-harm when overwhelmed.' And that's what I did as well. […] And so for years, when I was also self-harming due to being overwhelmed, I just thought that that's just how I am. That's just my mental health issues. But now, yeah, I realise it's completely triggered by things that are not autism-friendly, or too much masking. And yet, yeah, there isn't anything in the questionnaires. (Skye, woman, aged 36)*

**Subtheme 2.3: Daily life skills and repetitive behaviours.** Further experiences relevant to autistic women that have been identified as missing from the questionnaires include understanding how they navigate daily life and the significant role that repetitive behaviours and interests play into it, influencing aspects such as diet, self-care, and overall well-being.

*They don't ask anything about, for instance, about foods or restricted diet. They don't ask about rituals or like, at any point, repetitive activities or anything like that, or like stimming or anything. (Aubrey, woman, aged 48)*

The RRBIs highlighted most repeatedly by participants as missing were around eating habits. One participant mentioned that their eating habits were what first indicated to professionals that they might be autistic, yet such behaviours were not included in any of these measures.

*Nothing about meltdowns/shutdowns, executive functioning difficulties, stimming/repetitive behaviours. For me, sensory sensitivities/ being 'texture funny' with food is one of the big things that tipped both OT (occupational therapists) and dietetics off about me being autistic. (Poppy, non-binary, aged 29)*

The inclusion of RRBIs was also highlighted as important in the context of the questionnaires being overly focused on social interactions, an area where autistic women and those socialised as female can often mask, and thus potentially avoid detection because the measures do not account for it. Participants noted that RRBIs could potentially serve as compensatory strategies for distress or burnout related to masking, making their inclusion pivotal for detecting those who mask.

*[…] anything about repetitive behaviours [would be beneficial to include]. […] I feel like having that kind of big focus on the social situations means that actually you're probably missing quite a lot of behaviours that should be used to compensate for masking so intensely, like stimming, like shutdowns, things like that. (Danielle, woman, aged 25)*

**Subtheme 2.4: Navigating relationships and vulnerability.** Participants strongly emphasised the need for questions about how autistic women and individuals socialised as female navigate relationships, as they can face distinct challenges. They may often struggle more with maintaining friendships than initiating them and frequently experience shifts in and out of social circles without clear reasons. Helen pointed out how they may also tend to develop shallower connections, struggling to connect on a deeper level.

*I wish that [questionnaires] had, 'oh, you can make acquaintances, but … you can't quite make true friends' or, oh, 'you have few friends and you went in and out of the social groups' because like I heard that's very common with autistic women. (Helen, woman, aged 25)*

Robin further underscored the unique challenges faced by women and those socialised as female, particularly the heightened pressure to fit in and the complexities of navigating female relationships, which could be included in autism measures to enhance their relevance.

*I think there is among women [...] this perceived need to fit in perhaps a bit more [...] there's these weird like social politics that are like not as easy to understand. And I just don't know if that's, I don't think that's quite the same level of complexity with male friendships […] (Robin, woman and non-binary, aged 30)*

Vulnerability in relationships was also highlighted as a critical area to cover in questionnaires. Participants stressed the need for questions addressing their strong desire for acceptance, fitting in, and forming friendships and partnerships, amidst their uncertainty about navigating these relationships, which can leave them vulnerable to being taken advantage of or abuse. Participants suggested that including these questions could help autistic women in recognising how this vulnerability may relate to autism. Sarah elaborated on their own experiences, highlighting the significance of integrating such questions into measures.

*And I feel like the fact that I take what people say at face value, and I don't think people are trying to like hurt you or like be deceptive, because I'm not, it makes you vulnerable for those things. So I think [if] there are questions or conversations about that, it would be super helpful. And also like, like could [be] life-saving as well, right? Like, important for your safety and well-being, in addition to helping you realise different ways, like being autistic impacts your life and your lived experience. (Sarah, genderqueer, aged 36)*

**Subtheme 2.5: Strong empathy and fairness.** Participants emphasised the need for questions about empathy and its connection to strong feelings against unfairness or injustice. They noted that empathy in autistic individuals varies widely, with some exhibiting hyper-empathy and others hypo-empathy. Robin explained that autistic women, in particular, can be often at the hyper-empathic end of the spectrum, which is not reflected in the questionnaires.

*I think that empathy is massive. Like this bit about being like ridiculously empathetic. It goes against like the classic stereotypes, doesn't it? Of autistics not being empathetic. And again, I think is more of a- not saying that autistic men aren't empathetic, but I do think autistic women are particularly. And that's not captured in... I'm not sure it's captured in assessment very much, really. (Robin, woman and non-binary, aged 30)*

Heightened empathy was frequently linked to a strong sense of justice and fairness, with participants noting that autistic individuals can often experience these feelings more deeply and intensely than neurotypical people. They described a

reluctance to compromise in the face of injustice, even at the risk of social disapproval or isolation. Emilia underscored the profound connection of these traits to autism:

> And so I was really surprised that there weren't questions about that [justice and fairness] because that's such a fundamental part of being autistic, I think. (Emilia, woman, aged 32)

## Discussion

The objective of this study was to qualitatively explore the relevance of three frequently used autism questionnaires, namely the AQ-10, RAADS-14, and BAPQ, to the experiences of autistic women, while also incorporating additional perspectives from AFAB gender-diverse individuals. Drawing on an RTA of in-depth interviews, focus groups, and online responses from participants of a wide range of ethnicities, ages, and diagnostic journeys, the findings suggest that these questionnaires may not fully capture the experiences of autistic women and those socialised as female. Broader issues with the questionnaires' items were also identified, suggesting that these tools may lack relevance for autistic people more generally. The findings also indicated significant gaps in the questionnaires, indicating that key aspects of the experiences of autistic women and AFAB gender-diverse individuals might be missing. These results suggest unsatisfactory content validity of these measures in assessing autism among women and those socialised as female, emphasising the need for their refinement to reflect contemporary understandings of autism, particularly the impact of gendered experiences and expectations on the presentation of autistic traits.

### Irrelevant and missing concepts in autism questionnaires for autistic women and those socialised as female

Participants often described the three self-report questionnaires as reflecting a narrow and stereotyped portrayal of autistic traits, primarily based on behaviours typically observed in autistic boys, which may not apply to adult women – a finding reflected in previous research conducted with autistic women on the AQ-50 and diagnostic tools [65,67,68]. While many participants felt the measures reinforced a 'male autism' stereotype, some cautioned against attributing particular sets of autistic traits strictly to gender. Instead, they suggested distinguishing between more external, easily observable characteristics, and more internal autistic traits that may be less apparent through observation. This perspective partially aligns with research as perceived levels of 'atypical' autistic behaviours or perceived difficulties in females have been reported to vary by observer (for instance, teacher, parent, clinician), suggesting they may appear more or less salient depending on the environment [91,92]. For example, unlike parent reports, teachers tend to report more disruptive or 'atypical' behaviours in autistic boys than in autistic girls [91,92]. Although findings consistent with subtler variations of autistic traits are indeed more frequently observed in women and girls on average, conceptualising autism in terms of 'male' versus 'female' traits may risk being exclusionary for autistic people of other genders with this presentation, and may assume that all autistic women share the same presentation of autistic traits. Some participants' suggestions to move away from gendered frameworks of autistic traits highlight the need for research to explore alternative conceptualisations in collaboration with autistic community, and to consider how to avoid introduction of potentially new biases. Given these considerations, to ensure the relevance of measures to autistic women and those socialised as female, it may be therefore more advisable to focus on developing questionnaires that capture a broad range of autistic traits, especially at the behavioural exemplars levels, accommodating the full spectrum of presentations, rather than creating gender-specific versions.

In relation to the measures capturing a 'male autism' stereotype, participants in particular highlighted the lack of relevance of questions about passionate interests. The behavioural exemplars accompanying these questions, such as 'collecting information about categories of things, e.g., types of car, bird, train, plant etc.' (from AQ-10) did not resonate with autistic women and those socialised as female, who may have more 'normative' or relational interests. This reflects findings from the literature [46,47,68,93], which report that the interests of autistic women often align with those typically

seen in non-autistic women and girls, such as an affinity for books, art, animals, or people, rather than the mechanical or technology topics more commonly endorsed by autistic males. To better capture the passions of autistic women and those socialised as female, questionnaires should therefore include examples that reflect a broader range of interests. In line with participants' recommendations to emphasise the internal experiences of autism, we additionally suggest including questions that address not only the content of interests but also the quality and style of engagement with them. Specifically, emphasis could be placed on hyper-focus and the achievement of 'flow states,' either for intrinsic pleasure or as a strategy for self-regulation [94]. Such a monotropic style of engagement has been integrated into, for instance, the Monotropism Questionnaire [95]. Participants emphasised the importance of more relevant questions pertaining to their passions, given that these questionnaires were perceived as disproportionately influenced by autistic socio-communication traits, where their score may not be a true reflection of these traits due to masking and more nuanced socio-communication differences not being captured in the measures. This is particularly significant because participants reported engaging in RRBIs, including not only autistic passions but also stimming, and repetitive behaviours affecting daily life skills such as eating habits, as compensatory mechanisms for managing distress or burnout associated with masking. A recent study found that more frequent engagement with autistic passions, and greater distress when disengaging from them, predicted higher levels of masking [96]. It was suggested that autistic passions may possibly function as a means of recovery from the strain of masking, though this association requires further investigation [96]; nevertheless, the narratives of participants in our study provide support for this possibility. Notably, engagement with RRBIs was identified as lacking in the questionnaires, with only the AQ-10 addressing autistic passions, albeit in a manner that was not relevant to participants. Consequently, according to Haynes [1], such item construction can therefore lead to aggregate scores that overrepresent the socio-communication domain of autism, particularly in ways that fail to reflect autistic women's experiences, and underrepresent RRBIs, ultimately indicating less than satisfactory content validity of these measures for autistic women and those socialised as female. In turn, this may affect the screening process and influence who is ultimately referred for diagnosis.

The emphasis on observable autistic behaviours in questionnaires was considered problematic, which might possibly be tied to the historical assessment of autism in young children based on their external behaviours. This focus neglects the reality that autistic women are more likely to mask their autistic traits than autistic men [31,97,98], largely due to increased pressure to conform to gender and cultural expectations women may face – a finding supported by previous accounts of autistic women [37,99]. Participants emphasised that the ability to perform neurotypical behaviours, often as a result of gendered socialisation into performing corresponding gender roles, should not negate one's autistic identity, as these skills are not innate but learnt and maintained at considerable personal cost, with the impacts of masking well-documented in the literature [100,101]. Participants therefore underscored the need for questions that address the internal experiences of autism, to counteract the limitations of relying solely on observable behaviours, which will help to ensure that individuals who mask their autistic traits are not overlooked. Participants specifically recommended the inclusion of a dedicated section on masking within these measures to enhance their relevance to the experiences of autistic women and those socialised as female, mirroring similar recommendations for diagnostic assessments [65]. Indeed, masking has been recognised in the literature as a barrier in accessing diagnosis for autistic women [77], and the absence of questions addressing this issue may exacerbate these barriers. Encouragingly, recent instruments have begun to incorporate these considerations, as seen in the work of Ratto et al. [102], Hechler et al. [103], Brown et al. [70], Groen et al. [104], and English et al. [105]. Participants further emphasised that such a section should capture the experiences of individuals who may not be consciously aware of their masking behaviours, a pattern consistent with reports from other autistic people [106]. For example, it could include items about the internal experiences of autism, the functions of specific autistic traits, and a focus on respondents' preferences rather than their behaviours. For example, Hechler et al. [103] addressed this in the revised Comprehensive Autistic Trait Inventory (CATI) by reformulating items to require less self-insight and adding items for those who may not be fully aware of their own behaviours. Importantly, the inclusion of items on masking should be informed by further research

into the construct of autistic masking itself [11,45]. Moreover, as masking may not be autism-specific [42,43] but a form of broader human tendencies for impression management [44], such items must be developed with sufficient nuance to capture autism-specific aspects. As its conceptualisation continues to evolve and while recognising that not all autistic people engage in masking, or may do so sparingly [107], the inclusion of masking-related items may be best positioned as supplementary. In this capacity, it could serve to complement responses on core traits and help inform referral for diagnostic assessment in cases where individuals score below standard screening thresholds.

In addition to masking and RRBIs, participants identified several important concepts missing from these measures that are crucial in measuring autism in a way that is relevant to autistic women and individuals socialised as female: navigating relationships and vulnerability, as well as strong empathy and a strong sense of fairness. Although the questionnaires heavily focus on socio-communication traits, they fail to capture the nuanced social communication traits specific to their lived experience. These include a higher motivation for friendships, challenges in maintaining them, developing shallower connections, struggling to connect on a deeper level, and experiencing heightened pressure to fit in, with those accounts being supported by the literature [33–37]. The pressure to fit in and the need to belong have made some participants vulnerable to being taken advantage of or abuse. Some participants stressed the value of including statements on vulnerability as this could help autistic women and those socialised as female recognise how it relates to autism, potentially offering life-saving insights. This need also emerged in a study on autistic women's experiences of diagnostic assessments, where asking questions about sexual abuse and exploitation was deemed mandatory [65]. Given that 60% to 90% of autistic women are reported to have experienced serious sexual or domestic violence, with autism appearing to be a vulnerability factor [108], these conversations are critical. We recommend that such sensitive topics are more appropriately addressed within diagnostic assessments or post-diagnostic support, although we recognise that accessibility of the latter remains largely limited within the UK system [109]. While understanding how autistic traits may increase the risk of experiences such as abuse or victimisation is vital, it may go beyond the intended scope of screening tools focused solely on identifying autistic traits. Participants also highlighted the need for questions about empathy and concern for fairness, with the accounts of autistic women having a strong sense of justice and hyper-empathy being reported in the literature [37,110,111]. Some participants reported that autistic women may be particularly more hyper-empathetic than autistic males, reflecting the need for such statements in the measures. Clinicians have similarly noted heightened emotional empathy, accompanied by challenges in cognitive empathy, as characteristic of autistic women and considered the assessment of these traits clinically useful [112], supporting the inclusion of such items in questionnaires. Although autistic people of all genders have reported experiencing hyper-empathy [113], some research indicates that autistic women may have higher emotional than cognitive empathy when compared to autistic males [114]. Emotional empathy has been reported in some studies to be heightened in autistic people in comparison to non-autistic people [115] but those findings are mixed, with other studies indicating the same level of emotional empathy in autistic and non-autistic individuals [116,117]. Despite the mixed findings, the inclusion of well-designed items on emotional empathy may enable more accurate investigation of this construct. The need for such statements has been similarly recognised in the recent revision of CATI, which now includes item addressing empathy overload [103]. Overall, several aspects of the autism construct were identified by our participants as important but overlooked in these measures, suggesting their poor content validity for this population, as perceived by autistic women and those socialised as female. The inclusion of these additional constructs warrants further investigation with clinicians and researchers, in collaboration with the autistic community, to determine whether they are best incorporated into screening measures, diagnostic assessments, or instruments capturing broader autistic experience, as their inclusion should be guided by the intended purpose of the measure and may not be suitable in all contexts.

## Broader issues with the questionnaires

Broader issues with questionnaire items were identified that may apply to the autistic population in general, though this requires further investigation as experiences of autistic people of other genders were not included in this study. Several aspects of these questionnaires were identified as non-autism friendly.

First, the measures were perceived as portraying autistic traits as deficits – a perspective that does not align with the neurodiversity paradigm, and which left some participants feeling judged or offended. This suggests a need for rephrasing statements to be less stigmatising, ideally in collaboration with autistic individuals (see Hechler et al. [103], and Ratto et al. [102] for examples) or developing new measures if their current conceptualisation of autism is deemed stigmatising [118].

Second, consistent with findings from the literature on measures used with autistic populations and autistic women [66,67,118], participants highlighted that some items were overly specific (particularly those pertaining to sensory sensitivities), while others were perceived as too general, lacking necessary context to account for variations in behaviour across different situations. The specificity of these statements could be enhanced by incorporating a broader range of examples (for examples see for instance Hechler et al. [103] and Ratto et al. [102]), while the lack of context could be addressed by clarifying situational variables, such as interactions with strangers versus close acquaintances (although the BAPQ has incorporated this using an asterisk with an explanation below the scale, this approach risks being overlooked), experiences in new versus familiar environments, indoor versus outdoor settings, and distinguishing between what someone does versus what they would prefer to do. Nicolaidis et al. [118] also suggested including prefaces or more thorough instructions to provide context.

Third, the response scales were seen as insufficiently precise, making it challenging for participants to differentiate between the various categories, which aligns with existing accounts of autistic people on response scale options used in survey measures [66,118]. However, some participants in this study found RAADS-14 response scale ('true now and when I was young,' 'true only now,' 'true only when I was younger,' 'never true') helpful in distinguishing masking behaviours at different life stages. To address this, scale developers could improve clarity by quantifying frequency in response options, such as 'occasionally (1-2 times per week),' and possibly pairing frequency with an intensity scale to indicate that even infrequent occurrences can be very intense and impactful on everyday functioning. Nicolaidis et al. [118] also suggested adding graphics to increase clarity of the response options.

Finally, concerns were also raised about the requirement for self-insight when responding to these questionnaires, which was previously noted by autistic women with the CAT-Q [67]. However, in this study, this issue was identified across all three measures. Participants felt that it was unfair to expect autistic individuals to accurately compare their behaviours and perceptions to those of neurotypical individuals, particularly when their understanding of what is 'typical' may be influenced by interactions with neurodivergent family and friends. Participants also emphasised the difficulty of understanding what the questions were truly asking, noting that significant knowledge of autism was often required. Many reported needing or wanting to consult close friends/relatives or their GP for clarification or confirmation, aligning with reports from autistic women in another study, who valued opportunities to clarify their responses due to confusing item wording [68]. This was further exacerbated by participants taking the statements literally, underscoring the need for broader examples or an instructional caveat explaining that the expression of specific traits can vary widely from person to person.

Importantly, these limitations highlight the risks of over-reliance on questionnaires and their cut-off scores in the evaluation of autistic traits, indicating that results from such measures should be regarded as preliminary. They also underscore the importance of supplementing questionnaires with qualitative approaches whenever possible, whether in research context or screening for autism in primary care settings. The inclusion of qualitative input was frequently regarded by participants in this and one other study [68], as necessary, and is consistent with Cook's recommendations for more holistic assessments [26]. Such methods may help to address the inherent constraints of self-report instruments, which must remain brief for routine use and therefore may not fully capture all the ways autistic traits can present. Follow-up discussions after screening tool completion could mitigate some of the issues highlighted by participants by allowing individuals to contextualise their responses and provide behavioural examples or experiences absent from questionnaire measures, while also enabling researchers or healthcare professionals to probe for aspects that respondents themselves may not recognise as related to autism. This is particularly important as clinicians have noted that autistic women may find it challenging to complete questionnaires and may require prompting, particularly when questions are vague, and suggested that this trait may be informative in differential diagnosis, although it may not be gender-specific [112]. At the same time,

qualitative assessments also present challenges and may not always be feasible in primary care settings when screening for autism and in research contexts. Healthcare professionals can be influenced by biases and constrained by systemic barriers, including limited time, insufficient knowledge in recognising both subtler autistic traits and autism more broadly, underfunding, and wider pressures on healthcare systems, hindering the collection of a comprehensive clinical picture [119–121]. These feasibility issues are even more pronounced in research contexts, where studies often involve thousands of participants and individual follow-up interviews are impractical. Accordingly, efforts should be directed towards addressing these limitations at the level of the measurement tools themselves, as standardised instruments remain the most practical option in terms of ease of use, cost, and time efficiency.

## Limitations

While qualitative studies do not aim for generalisability, the findings may not fully represent the diverse experiences of all autistic women and AFAB gender-diverse individuals. The sample was largely recruited through social media, which itself may be biased towards certain populations [122], and given the capacity of platform algorithms to shape public opinion [123], it remains unclear to what extent discourse on these sites may have influenced participants' views. Participation may also have been skewed towards individuals with a particular interest in the topic or stronger opinions, and thus may not accurately represent even the demographics of the specific platforms from which they were recruited [124]. Moreover, most participants were university-educated and likely had a pre-existing interest in research, partly due to recruitment through the Autistica Network and Prolific, platforms specifically designed for individuals wishing to participate in research. Future research in this area would benefit from exploring the experiences of autistic men, those socialised as male, or those who identify with more conventional presentation of autism, to better understand potential differences and similarities in their experiences and perceptions of these autism measures. Additionally, as participants were from various cultural backgrounds, cultural differences in how autism is understood may have influenced their responses. Future studies should consider more culturally homogeneous samples to ensure that (non)relevance of autism measure items is not influenced by cultural factors. Additionally, abbreviated versions of the Autism Spectrum Quotient-50 [69] and the Ritvo Autism Asperger Diagnostic Scale–Revised [85] were used to reduce participant burden. Consequently, some limitations of these measures may possibly not have been identified had the full versions been used, while other discussions specific to the full versions may have been missed.

The inclusion of individuals who self-identify as autistic was a strength, especially considering the underdiagnosis, misdiagnosis, and delayed diagnosis often experienced by women [75]. Our intention was to include perspectives from individuals potentially overlooked by current autism measures, recognising that limiting participation to formally diagnosed individuals could reinforce existing biases. However, this approach also presents limitations: self-identification may not have been independently verified and may have evolved over time. Future research could consider prioritising individuals who have maintained an autistic identity over a longer period of time, as a way to address the latter concern. Furthermore, while some studies screen self-identifying participants using self-report tools, we opted not to do so, due to concerns regarding potential gender bias in such tools. Reassuringly, growing evidence supports the inclusion of people self-identifying as autistic, with findings often aligning with those formally diagnosed [125,126], as was observed in this study.

## Conclusions

This study examined the perspectives of autistic women and gender-diverse individuals socialised as female on the AQ-10, RAADS-14, and BAPQ. The findings highlight the unsatisfactory content validity of these tools for assessing autism in women and those socialised as female, and possibly for autistic people more broadly. The insights from this study could benefit individuals who have been overlooked by current autism measures, particularly autistic women and those socialised as female, by providing guidance on how to improve the relevance of these questionnaires to their

experiences. It is hoped that future scale developers will consider these findings to make screening practices more inclusive for everyone.

## Supporting information

**S1 Table. Interview guide.**
(PDF)

**S2 Table. Demographic characteristics of participants (N = 22).**
(PDF)

## Acknowledgments

The authors would like to express their sincere gratitude to the autistic individuals who provided feedback on the study materials and the manuscript, as well as those who participated in the study for their valuable contributions and engaging discussions. We also extend our thanks to Dorota Ali for her valuable comments on the manuscript.

## Author contributions

**Conceptualization:** Nora Uglik-Marucha, Silia Vitoratou, Francesca Happé, Hannah Belcher.

**Data curation:** Nora Uglik-Marucha.

**Formal analysis:** Nora Uglik-Marucha.

**Investigation:** Nora Uglik-Marucha.

**Methodology:** Nora Uglik-Marucha, Francesca Happé, Hannah Belcher.

**Project administration:** Nora Uglik-Marucha, Serafina Show.

**Supervision:** Silia Vitoratou, Francesca Happé, Hannah Belcher.

**Visualization:** Nora Uglik-Marucha.

**Writing – original draft:** Nora Uglik-Marucha.

**Writing – review & editing:** Nora Uglik-Marucha, Serafina Show, Silia Vitoratou, Francesca Happé, Hannah Belcher.

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
