## [Decision Letter · Decision Letter 0]

21 Aug 2025

“I fit the category of the box, it just doesn't describe me well.” Exploring the perspectives of autistic women and gender-diverse individuals on self-report autism measures

PLOS ONE

Dear Dr. Uglik-Marucha,

Thank you for submitting your manuscript to PLOS ONE. After careful consideration, we feel that it has merit but does not fully meet PLOS ONE’s publication criteria as it currently stands. Therefore, we invite you to submit a revised version of the manuscript that addresses the points raised during the review process.

Your article has been reviewed by two experts. Both of them agreed that it enriches knowledge on diagnosing adult women on the autism spectrum. However, the reviewers also raised a number of comments, the inclusion of which could significantly improve the paper. From my perspective, the remarks concerning methodology (e.g., provide a clearer rationale for data saturation) are particularly important, as well as the interesting thread on the conceptualization of masking. There are also several valuable suggestions regarding the enrichment of the discussion.

We look forward to receiving your revised manuscript.

Kind regards,

Ewa Pisula

Academic Editor

PLOS ONE

Journal Requirements:

2. Please describe in your methods section how capacity to provide consent was determined for the participants in this study. Please also state whether your ethics committee or IRB approved this consent procedure. If you did not assess capacity to consent please briefly outline why this was not necessary in this case.

[SV and FH are partially funded by the NIHR Maudsley Biomedical Research Centre at South London and Maudsley NHS Foundation Trust and King’s College London. NUM is funded by the NIHR [Doctoral Fellowship (NIHR302618)]. The views expressed are those of the author(s) and not necessarily those of the NHS, the NIHR, or the Department of Health and Social Care. The funders had no role in study design, data collection and analysis, decision to publish, or preparation of the manuscript. There was no additional external funding received for this study.].

5. Please amend the manuscript submission data (via Edit Submission) to include author Nora Uglik-Marucha.

6. Please amend your authorship list in your manuscript file to include author Eleonora Uglik-Marucha.

Comments from the editorial office:

Upon internal evaluation of the reviews provided, we kindly request you to disregard the reviewer report provided by Reviewer 3. No amendments are required in response to reviewer 3’s comments

Additional Editor Comments:

Your article has been reviewed by two experts. Both of them agreed that it enriches knowledge on diagnosing adult women on the autism spectrum. However, the reviewers also raised a number of comments, the inclusion of which could significantly improve the paper. From my perspective, the remarks concerning methodology (e.g., provide a clearer rationale for data saturation) are particularly important, as well as the interesting thread on the conceptualization of masking. There are also several valuable suggestions regarding the enrichment of the discussion.

Reviewers' comments:

Reviewer's Responses to Questions

**Comments to the Author**

1. Is the manuscript technically sound, and do the data support the conclusions?

Reviewer #1: Yes

Reviewer #2: Partly

2. Has the statistical analysis been performed appropriately and rigorously?

Reviewer #1: Yes

Reviewer #2: N/A

3. Have the authors made all data underlying the findings in their manuscript fully available?

Reviewer #1: Yes

Reviewer #2: No

4. Is the manuscript presented in an intelligible fashion and written in standard English?

Reviewer #1: Yes

Reviewer #2: Yes

Reviewer #1: Thank you for the opportunity to review the manuscript entitled ““I fit the category of the box, it just doesn't describe me well.” Exploring the perspectives of autistic women and gender-diverse individuals on self-report autism measures”. Here are some suggestions that I hope will be interesting and helpful for the Authors:

1) First of all, I would like to highlight that I fully agree with your paper’s thesis and I find it very important for the community, both in terms of clinical practice and research. At the same time, one of the main issues of screening autism or measuring its traits is focusing only on quantitative psychological assessments which makes it reductionistic and may be one of the main issues in current clinical practice when it comes to autism. Therefore, although I understand that your paper focuses on quantitative research tools, I would suggest to expand the context of your Discussion on a very important aspect of qualitative interview (especially among adults), as no psychological measurement can surpass this way of gathering clinical information because it grasps way more context and aspects of one’s functioning. I think that this approach would be in line with your results, e.g., participants’ comments about lack of context or too narrow (or too broad) categories/items/statements presented in quantitative items. I see that you suggested that we need a quantitative measurement that addresses these issues but I wonder if that is actually possible to achieve.

2) On page 8, you comment that a proper quantitative measurement of autistic traits would focus on DSM-5 symptoms, monotropism, camouflaging, etc. I would suggest to discuss another important topic in a form of compensatory strategies (other than camouflaging), as these often make quantitative tools quite useless as they do not grasp these sort of aspects, especially among older adolescents and adults, regardless of their gender (e.g., some autistic individuals may state that they do not experience sensory sensitivity everyday because they have developed convenient and supportive strategies, not because they are not sensitive to sensory input, etc.).

3) On page 16, you described AQ-10 as “widely utilized in research, it has accumulated over 800 citations on Google Scholar and is commonly employed for participant inclusion and exclusion criteria in studies.” That is of course true although I would suggest adding an information about a growing literature regarding concerns about AQ-10’s reliability and validity issues, both in research and clinical practice (see: Bertrams, 2021; Taylor et al., 2020). I think that it would be important to point at this issue, especially in the context of your work.

4) Table 2 shows that 8 participants were AuDHDers and I wonder if you noticed any differences between AuDHD participants when compared to autistic ones? I think that it is possible that AuDHDers would vary in terms of their perception of understanding or identifying with the items presented. This information would be useful if added to the Participants section.

Reviewer #2: Thank you for inviting me to review the manuscript entitled “I fit the category of the box, it just doesn't describe me well.” Exploring the perspectives of autistic women and gender-diverse individuals on self-report autism measures”. The manuscript focuses on the important topic of content validity of autism measures, particularly for autistic women and gender-diverse individuals, using a qualitative design and reflexive thematic analysis. Overall, I find the manuscript to be well-written, well-structured, and importantly, sufficiently nuanced to avoid reinforcing gender stereotypes in autism.

Main comments:

The manuscript raises important issues regarding the content validity of commonly used autism questionnaires. However, it would benefit from a clearer distinction between the purpose of the included clinical measures and broader conceptualizations of autism. The introduction and methods focus on instruments recommended for autism screening and assessments. Still, in the discussion, the authors suggest poor content validity due to missing aspects of the autism construct, such as emotional empathy and fairness. Given that construct validity is tied to the instrument’s intended purpose, as stated in the introduction, I suggest expanding the discussion to differentiate between clinical utility and the broader goal of understanding autistic lived experience, which may not fully overlap.

Additionally, the authors report that the findings on emotional empathy in autism are mixed which seems to indicate that more research on the topic is needed before inclusion in autism screening questionnaires is considered.

The topic of masking is discussed throughout the manuscript. While masking is a common experience among autistic adults, it does not seem to be specific to autistic people, (e.g., overlaps with impression management in ADHD, social anxiety; see for example, Ai et al (2024) https://doi.org/10.1016/j.comppsych.2023.152434). In the following sentence, masking appears to be portrayed as a facet of autism rather than compensatory strategies: “Masking has been suggested to constitute subtler variations of autistic traits, though further research is needed, potentially contributing to the under-recognition of autism in females...” (p. 7). I suggest that the authors clarify their conceptualization of masking and discuss its presence in non-autistic populations. It would also be helpful to elaborate on how masking could be assessed within an autism screening instrument, given the concerns on its specificity stated above.

The recruitment channels are described, but the manuscript lacks information on how recruitment was conducted, specifically, how the study was advertised. This is important as it may influence participant self-selection. For example, could the recruitment procedure specifically have attracted those with certain/strong opinions regarding the topic which are then generalized to autistic women?

More information on data saturation and the rationale for the final sample size would strengthen the manuscript. Was data saturation evaluated during data collection? What guided the inclusion of the final sample, were stopping rules or other criteria used?

I also suggest that the authors briefly discuss the potential influence of online communities, social media, and recent research on participants’ responses. Given the increased interest in the topic it would be good to acknowledge this possibility.

Minor:

p. 19 – The reason for excluding 3 participants were vague. A reference is given, but that paper includes a range of issues. I suggest briefly summarizing the reasons for exclusion more specifically to improve transparency.

p. 6 – When reporting sex/gender differences findings on the narrow construct level, a broad description is given, “autistic females tend to exhibit better social communication skills than autistic males” (p. 6). Subsequently, broad level findings are described in more granular terms and split into social communication and interaction respectively, which makes the passage slightly confusing. Please revise for clarity.

p. 19 - In the sample, 72.7% are reported to have a diagnosis of autism – where diagnoses self-reported or verified through clinical documentation?

p. 27 – the name of subtheme 1.4: "Non-autistic friendly process" sounds like an uncommon phrasing. Since “autism-friendly” is used elsewhere, I suggest revising to “not autism-friendly process” or “non-autism-friendly process”.

RRBIs are suggested to be used as “compensatory mechanisms for masking” in both the results and discussion. This interpretation is not clear to me – could the authors elaborate on what is meant by this and provide examples?

**Do you want your identity to be public for this peer review?** For information about this choice, including consent withdrawal, please see our Privacy Policy

Reviewer #1: Yes**: ** Anna Pyszkowska

Reviewer #2: No

---

## [Author Response · Author response to Decision Letter 1]

11 Oct 2025

Dear Editor and Reviewers,

Please find below a summary of the reviewers’ comments along with our responses. The line references provided correspond to the version of the manuscript with tracked changes.

Editor Comments:

Your article has been reviewed by two experts. Both of them agreed that it enriches knowledge on diagnosing adult women on the autism spectrum. However, the reviewers also raised a number of comments, the inclusion of which could significantly improve the paper. From my perspective, the remarks concerning methodology (e.g., provide a clearer rationale for data saturation) are particularly important, as well as the interesting thread on the conceptualization of masking. There are also several valuable suggestions regarding the enrichment of the discussion.

We thank the two reviewers and the editor for these insightful comments and hope that our revisions address them satisfactorily. We look forward to your feedback on the changes made.

Reviewer #1:

Thank you for the opportunity to review the manuscript entitled ““I fit the category of the box, it just doesn't describe me well.” Exploring the perspectives of autistic women and gender-diverse individuals on self-report autism measures”. Here are some suggestions that I hope will be interesting and helpful for the Authors:

1. First of all, I would like to highlight that I fully agree with your paper’s thesis and I find it very important for the community, both in terms of clinical practice and research. At the same time, one of the main issues of screening autism or measuring its traits is focusing only on quantitative psychological assessments which makes it reductionistic and may be one of the main issues in current clinical practice when it comes to autism. Therefore, although I understand that your paper focuses on quantitative research tools, I would suggest to expand the context of your Discussion on a very important aspect of qualitative interview (especially among adults), as no psychological measurement can surpass this way of gathering clinical information because it grasps way more context and aspects of one’s functioning. I think that this approach would be in line with your results, e.g., participants’ comments about lack of context or too narrow (or too broad) categories/items/statements presented in quantitative items. I see that you suggested that we need a quantitative measurement that addresses these issues but I wonder if that is actually possible to achieve.

We thank the reviewer for this comment, which has enabled us to enrich the discussion. We agree that no quantitative measure can substitute for a qualitative interview, and that the limitations of the measures highlighted by participants illustrate the risks of over-reliance on questionnaires and their cut-off scores. In the revised discussion, we now emphasise that, the completion of quantitative measures should be supplemented with qualitative methods, while also acknowledging the limitations of such approaches. We further note the importance of addressing the highlighted issues at the level of measurement itself, given the practical advantages of standardised instruments.

Lines 1160 to 1192: “Importantly, these limitations highlight the risks of over-reliance on questionnaires and their cut-off scores in the evaluation of autistic traits, indicating that results from such measures should be regarded as preliminary. They also underscore the importance of supplementing questionnaires with qualitative approaches whenever possible, whether in research context or screening for autism in primary care settings. The inclusion of qualitative input was frequently regarded by participants in this and one other study [68], as necessary, and is consistent with Cook’s recommendations for more holistic assessments [26]. Such methods may help to address the inherent constraints of self-report instruments, which must remain brief for routine use and therefore may not fully capture all the ways autistic traits can present. Follow-up discussions after screening tool completion could mitigate some of the issues highlighted by participants by allowing individuals to contextualise their responses and provide behavioural examples or experiences absent from questionnaire measures, while also enabling researchers or healthcare professionals to probe for aspects that respondents themselves may not recognise as related to autism. This is particularly important as clinicians have noted that autistic women may find it challenging to complete questionnaires and may require prompting, particularly when questions are vague, and suggested that this trait may be informative in differential diagnosis, although it may not be gender-specific [112]. At the same time, qualitative assessments also present challenges and may not always be feasible in primary care settings when screening for autism and in research contexts. Healthcare professionals can be influenced by biases and constrained by systemic barriers, including limited time, insufficient knowledge in recognising both subtler autistic traits and autism more broadly, underfunding, and wider pressures on healthcare systems, hindering the collection of a comprehensive clinical picture [119–121]. These feasibility issues are even more pronounced in research contexts, where studies often involve thousands of participants and individual follow-up interviews are impractical. Accordingly, efforts should be directed towards addressing these limitations at the level of the measurement tools themselves, as standardised instruments remain the most practical option in terms of ease of use, cost, and time efficiency.”

2. On page 8, you comment that a proper quantitative measurement of autistic traits would focus on DSM-5 symptoms, monotropism, camouflaging, etc. I would suggest to discuss another important topic in a form of compensatory strategies (other than camouflaging), as these often make quantitative tools quite useless as they do not grasp these sort of aspects, especially among older adolescents and adults, regardless of their gender (e.g., some autistic individuals may state that they do not experience sensory sensitivity everyday because they have developed convenient and supportive strategies, not because they are not sensitive to sensory input, etc.).

We thank the reviewer for this suggestion. While we fully agree with the point raised and recognise the value of including such strategies in questionnaires, we chose to prioritise the suggestions that emerged from participant discussions. We did not create a subtheme on this topic due to the scarcity of discussion on it. However, we are currently analysing qualitative data from a Delphi study with autistic women, clinicians, and researchers on the relevance of existing questionnaire items to autistic women. It is very possible that this suggestion has arisen there, and we will address compensatory strategies in the context of that study.

3. On page 16, you described AQ-10 as “widely utilized in research, it has accumulated over 800 citations on Google Scholar and is commonly employed for participant inclusion and exclusion criteria in studies.” That is of course true although I would suggest adding an information about a growing literature regarding concerns about AQ-10’s reliability and validity issues, both in research and clinical practice (see: Bertrams, 2021; Taylor et al., 2020). I think that it would be important to point at this issue, especially in the context of your work.

Thank you, we have now incorporated this in our manuscript.

Lines 416-418: “Nevertheless, despite its extensive use in both research and clinical settings, psychometric concerns have been raised regarding the measure [86,87], warranting further evaluation and caution in its continued use.”

4. Table 2 shows that 8 participants were AuDHDers and I wonder if you noticed any differences between AuDHD participants when compared to autistic ones? I think that it is possible that AuDHDers would vary in terms of their perception of understanding or identifying with the items presented. This information would be useful if added to the Participants section.

We thank the reviewer for this comment. Two participants introduced brief discussions about the overlap between autism and ADHD, highlighting that some autistic traits may be overshadowed by ADHD traits and that completing autism measures requires self-insight to distinguish between one’s autism and ADHD traits, which can be challenging. While we recognise the importance of this issue, we believe it warrants a separate investigation to provide the depth and nuance it deserves. In this paper, we chose to focus specifically on gender, namely, the relevance of autism measures to women and AFAB gender-diverse individuals, rather than introducing multiple but surface-level discussions of how completion of these measures may intersect with other conditions or identities, such as ADHD or cultural identities (which were also briefly discussed). The manuscript is already extensive in scope, and we felt it was important to maintain a clear and focused narrative. However, we are grateful for the suggestion and will keep it in mind for future studies.

Reviewer #2:

Thank you for inviting me to review the manuscript entitled “I fit the category of the box, it just doesn't describe me well.” Exploring the perspectives of autistic women and gender-diverse individuals on self-report autism measures”. The manuscript focuses on the important topic of content validity of autism measures, particularly for autistic women and gender-diverse individuals, using a qualitative design and reflexive thematic analysis. Overall, I find the manuscript to be well-written, well-structured, and importantly, sufficiently nuanced to avoid reinforcing gender stereotypes in autism.

1. The manuscript raises important issues regarding the content validity of commonly used autism questionnaires. However, it would benefit from a clearer distinction between the purpose of the included clinical measures and broader conceptualizations of autism. The introduction and methods focus on instruments recommended for autism screening and assessments. Still, in the discussion, the authors suggest poor content validity due to missing aspects of the autism construct, such as emotional empathy and fairness. Given that construct validity is tied to the instrument’s intended purpose, as stated in the introduction, I suggest expanding the discussion to differentiate between clinical utility and the broader goal of understanding autistic lived experience, which may not fully overlap.

We thank the reviewer for this discussion. We wish to clarify that the poor content validity of these measures was not solely attributable to the omission of experiences identified as important by autistic women and those socialised as female but was supported by several factors discussed in the manuscript. In line with the reviewer’s suggestion, we emphasise that the omission of these experiences suggests poor content validity for this population, specifically as perceived by autistic women and those socialised as female; had the study involved clinicians or researchers, the inclusion of these experiences might have been evaluated differently, with reference to the intended function of the measures. We therefore note that further research with clinicians and researchers is needed to determine whether such constructs are best incorporated into screening measures, diagnostic assessments, or broader lived-experience instruments, given that their assessment may not be appropriate in all contexts.

Lines 1091 to 1098: “Overall, several aspects of the autism construct were identified by our participants as important but overlooked in these measures, suggesting their poor content validity for this population, as perceived by autistic women and those socialised as female. The inclusion of these additional constructs warrants further investigation with clinicians and researchers, in collaboration with the autistic community, to determine whether they are best incorporated into screening measures, diagnostic assessments, or instruments capturing broader autistic experience, as their inclusion should be guided by the intended purpose of the measure and may not be suitable in all contexts.”

2. Additionally, the authors report that the findings on emotional empathy in autism are mixed which seems to indicate that more research on the topic is needed before inclusion in autism screening questionnaires is considered.

We thank the reviewers for this comment. While we agree that further research is needed, clinicians have identified the evaluation of emotional empathy as particularly useful in the assessment of autism in women (Cumin et al., 2021), thereby warranting the implementation of such items. We have now added this point to the manuscript. The inclusion of well-constructed items on emotional empathy may also strengthen its measurement, allowing more accurate investigation in autistic people and supporting further research in this area, which we have further emphasised. Additionally, in response to point 1, we also note that additional research is required to determine whether such constructs should be incorporated into specific measures (see comment above).

Lines 1072 to 1075: “Clinicians have similarly noted heightened emotional empathy, accompanied by difficulties in cognitive empathy, as characteristic of autistic women and considered the assessment of these traits clinically useful for diagnosis [109], supporting the inclusion of such items in questionnaires.”

Lines 1080 to 1089: “Despite the mixed findings, the inclusion of well-designed items on emotional empathy may enable more accurate investigation of this construct.”

References:

Cumin, J., Pelaez, S., & Mottron, L. (2022). Positive and differential diagnosis of autism in verbal women of typical intelligence: A Delphi study. Autism : the international journal of research and practice, 26(5), 1153–1164. https://doi.org/10.1177/13623613211042719

3. The topic of masking is discussed throughout the manuscript. While masking is a common experience among autistic adults, it does not seem to be specific to autistic people, (e.g., overlaps with impression management in ADHD, social anxiety; see for example, Ai et al (2024) https://doi.org/10.1016/j.comppsych.2023.152434). In the following sentence, masking appears to be portrayed as a facet of autism rather than compensatory strategies: “Masking has been suggested to constitute subtler variations of autistic traits, though further research is needed, potentially contributing to the under-recognition of autism in females...” (p. 7). I suggest that the authors clarify their conceptualization of masking and discuss its presence in non-autistic populations. It would also be helpful to elaborate on how masking could be assessed within an autism screening instrument, given the concerns on its specificity stated above.

Thank you for highlighting this. We agree with the reviewer that the sentence concerning masking required clearer wording. We have removed the phrase “masking has been suggested to constitute subtler variations of autistic traits”, rephrased the section and incorporated the reviewer’s additional points.

Lines 146 to 152: “Better social communication skills could also be explained by masking, that is, the use of conscious or unconscious strategies to change social behaviour so that autistic differences are less apparent [38,39] or compensated for [40], in order to facilitate navigation of predominantly neurotypical social contexts [41]. Although aspects of masking likely extend to non-autistic people as well [42,43], for autistic people it can come with distinct motives and costs [44], and may be modulated by gender [45], potentially contributing to the under-recognition of autism in females.”

Lines 1020 to 1041: “Importantly, the inclusion of items on masking should be informed by further research into the construct of autistic masking itself [11,45]. Moreover, as masking may not be autism-specific [42,43] but a form of broader human tendencies for impression management [44], such items must be developed with sufficient nuance to c

---

## [Decision Letter · Decision Letter 1]

11 Nov 2025

“I fit the category of the box, it just doesn't describe me well.” Exploring the perspectives of autistic women and gender-diverse individuals on self-report autism measures

PONE-D-25-24929R1

Dear Dr. Nora Uglik-Marucha,

We’re pleased to inform you that your manuscript has been judged scientifically suitable for publication and will be formally accepted for publication once it meets all outstanding technical requirements.

Kind regards,

Claudia Brogna

Academic Editor

PLOS ONE

Additional Editor Comments (optional):

Reviewers' comments:

Reviewer's Responses to Questions

**Comments to the Author**

Reviewer #1: All comments have been addressed

Reviewer #2: All comments have been addressed

2. Is the manuscript technically sound, and do the data support the conclusions?

Reviewer #1: Yes

Reviewer #2: Yes

3. Has the statistical analysis been performed appropriately and rigorously?

Reviewer #1: Yes

Reviewer #2: N/A

4. Have the authors made all data underlying the findings in their manuscript fully available?

Reviewer #1: Yes

Reviewer #2: No

5. Is the manuscript presented in an intelligible fashion and written in standard English?

Reviewer #1: Yes

Reviewer #2: Yes

Reviewer #1: I thank the Authors for their work on the manuscript. I believe that this version of the paper is sound and sufficient for publication.

Reviewer #2: I feel that the authors have responded well to my comments, and that the manuscript is ready for publication.

**Do you want your identity to be public for this peer review?** For information about this choice, including consent withdrawal, please see our Privacy Policy

Reviewer #1: **Yes: ** Anna Pyszkowska

Reviewer #2: **Yes: ** Karl Lundin Remnélius

---

## [Editor Report · Acceptance letter]

PONE-D-25-24929R1

PLOS ONE

Dear Dr. Uglik-Marucha,

I'm pleased to inform you that your manuscript has been deemed suitable for publication in PLOS ONE. Congratulations! Your manuscript is now being handed over to our production team.

Kind regards,

on behalf of

Dr. Claudia Brogna

Academic Editor

PLOS ONE